# Role of UDP-Sugar Receptor P2Y_14_ in Murine Osteoblasts

**DOI:** 10.3390/ijms21082747

**Published:** 2020-04-15

**Authors:** Nicholas Mikolajewicz, Svetlana V. Komarova

**Affiliations:** 1Shriners Hospital for Children, Montreal, QC H4A 0A9, Canada; 2Faculty of Dentistry, McGill University, Montreal, QC H3A 1G1, Canada

**Keywords:** P2Y_14_, *Gpr105*, osteoblasts, bone, purinergic receptors

## Abstract

The purinergic (P2) receptor P2Y_14_ is the only P2 receptor that is stimulated by uridine diphosphate (UDP)-sugars and its role in bone formation is unknown. We confirmed P2Y_14_ expression in primary murine osteoblasts (CB-Ob) and the C2C12-BMP2 osteoblastic cell line (C2-Ob). UDP-glucose (UDPG) had undiscernible effects on cAMP levels, however, induced dose-dependent elevations in the cytosolic free calcium concentration ([Ca^2+^]_i_) in CB-Ob, but not C2-Ob cells. To antagonize the P2Y_14_ function, we used the P2Y_14_ inhibitor PPTN or generated CRISPR-Cas9-mediated P2Y_14_ knockout C2-Ob clones (Y14_KO_). P2Y_14_ inhibition facilitated calcium signalling and altered basal cAMP levels in both models of osteoblasts. Importantly, P2Y_14_ inhibition augmented Ca^2+^ signalling in response to ATP, ADP and mechanical stimulation. P2Y_14_ knockout or inhibition reduced osteoblast proliferation and decreased ERK1/2 phosphorylation and increased AMPKα phosphorylation. During in vitro osteogenic differentiation, P2Y_14_ inhibition modulated the timing of osteogenic gene expression, collagen deposition, and mineralization, but did not significantly affect differentiation status by day 28. Of interest, while *P2ry14^-/-^* mice from the International Mouse Phenotyping Consortium were similar to wild-type controls in bone mineral density, their tibia length was significantly increased. We conclude that P2Y_14_ in osteoblasts reduces cell responsiveness to mechanical stimulation and mechanotransductive signalling and modulates osteoblast differentiation.

## 1. Introduction

Purinergic (P2) receptors are a primitive family of extracellular nucleotide-sensing receptors that emerged early in our evolutionary history [1]. To date, 15 subtypes have been cloned and they are broadly classified as ionotropic P2X receptors (P2X_1–7_) and G-protein coupled P2Y receptors (P2Y_1, 2, 4, 6, 11–14_). P2X receptors are cation channels that are selectively activated by ATP while P2Y receptors exhibit a range of subtype-specific sensitivities to ATP, ADP, UTP, and UDP, allowing for context-specific responses. Few cells express each P2 receptor subtypes and rather tissue-specific patterns of P2 receptors allow for diverse functions such as mechano-sensation in the skin [2], bladder [3], and gastrointestinal tract [4], as well as regulation of blood flow in vascular endothelia [5], osmoregulation in the kidney [6], mucociliary clearance in the airway [7], and modulation of remodelling in bone [8].

The P2Y_14_ receptor is unique among the P2 receptor family because it is the only member that selectively responds to UDP-sugars including UDP-glucose (UDPG) and UDP-galactose [9]. Just as adenine nucleotides were first described for their intracellular role in cellular energetics [10], UDP-sugars are primarily known as sugar donors for glycosylation reactions in the endoplasmic reticulum and Golgi apparatus [11,12]. Among these UDP-glucose serves as an intermediate in several biosynthetic pathways, including synthesis of polysaccharides such as glycogen, lipopolysaccharides and glycosphingolipids [13]. The extracellular function of UDP-sugars was shown to follow their release through Ca^2+^-regulated exocytosis into the extracellular space, where they are enzymatically stable and capable of stimulating the P2Y_14_ receptor [14,15,16,17,18]. 

In bone, mechanical loading leads to the rapid release of extracellular nucleotides through vesicular and conductive pathways [19], as well as through reversible membrane disruption [20], thereby positioning P2 receptor signalling upstream of the mechano-adaptive skeletal response. In vitro studies have demonstrated that certain P2 receptors promote anabolic responses in osteoblasts (e.g., P2X_5_ [21], P2Y_1_ [22,23,24,25], P2Y_12_ [26,27], P2Y_13_ [28,29]) while others have catabolic effects through increased osteoclast activity (e.g., P2X_2_ [30], P2X_4_ [31,32,33], P2X_5_ [34], P2X_7_ [35,36,37,38]), thereby framing the P2 receptor family as a complex signalling network in the skeletal system. Skeletal phenotypes have been observed in nearly every P2 knockout strain established, including increased bone mineral density (BMD) or protection against bone loss in *P2rx2^-/-^* [39], *P2rx5^-/-^* [34], *P2ry2^-/-^* [40,41], *P2ry6^-/-^* [40], *P2ry12^-/-^* [42], or decreased BMD in *P2rx7^-/-^* [43] and *P2ry1^-/-^* [40] mouse models. The role of P2 receptors in mechanotransduction is further highlighted by reports of mechanically-loaded *P2ry13^-/-^* mice being associated with enhanced osteogenesis compared to wild-type controls [44,45] while mechanically-loaded *P2rx7^-/-^* mice showed reduced mechanosensitivity [46]. P2Y_14_ is evolutionarily-related to G_i_-coupled P2Y_12_ and P2Y_13_ receptors so it may be expected to have similar skeletal functions [47,48]. Outside of bone, P2Y14 has been implicated in modulating immune function, skeletal and smooth muscle contractions and glucose metabolism [14], thus suggesting context-specific roles for this receptor. To date, the skeletal phenotype of P2Y_14_-deficient mice has not been characterized, nor has the function of P2Y_14_ in bone-forming osteoblasts or bone-embedded osteocytes been described. 

The goal of this study was to determine if functional P2Y_14_ is expressed in murine osteoblasts. We investigated P2Y_14_ coupling to adenylyl cyclase (AC) inhibition and intracellular free calcium ([Ca^2+^]_i_) mobilization in compact bone-derived primary osteoblasts (CB-Ob) and BMP2-expressing C2C12 osteoblastic cells (C2-Ob). UDPG was used to stimulate P2Y_14_, while P2Y_14_ inhibition was achieved by pharmacological (P2Y_14_ antagonist PPTN [49]) and genetic (CRISPR-Cas9) interventions. We examined the contribution of P2Y_14_ to purinergic mechanotransduction, osteoblast proliferation and differentiation in vitro, and in vivo skeletal phenotype of P2Y_14_-deficient mice was acquired from a publicly available International Mouse Phenotyping Consortium (IMPC). 

## 2. Results

### 2.1. Functional P2Y_14_ is Expressed in Primary Murine Osteoblasts

P2Y_14_ gene expression has been previously reported in all bone-residing cell types, including hematopoietic (HSC; human, murine) and mesenchymal (MSC; human) stem cells, osteoblasts (human, murine, rat), osteocytes (murine) and osteoclasts (murine) (Table 1). At the protein-level, P2Y_14_ expression has been verified in murine HSCs and osteoclasts (Table 1). Using immunofluorescence, we found that compact bone-derived murine osteoblasts (CB-Ob) expressed P2Y_14_ (Figure 1a). To investigate P2Y_14_-mediated signalling, Fura2-loaded CB-Ob cells were stimulated with UDPG and [Ca^2+^]_i_ elevations were recorded (Figure 1b). UDPG-induced [Ca^2+^]_i_ elevations in primary murine osteoblasts with an estimated EC_50_ of 5.3 µM (95% CI: 2.5 to 8.0). In the presence of P2Y_14_ inhibitor PPTN (Y14_inh_), UDPG-stimulated calcium responses were potentiated (*p* = 0.04, 2-way ANOVA), however, there was no discernable dose-dependency. We next examined cAMP signalling and observed that forskolin (FK)-stimulated cAMP production was not reversed by UDPG-mediated P2Y_14_ stimulation, however inhibiting P2Y_14_ led to significant increases in basal cAMP levels (*p* < 0.01) (Figure 1c). Finally, UDPG-stimulated Y14_inh_-sensitive stress-fibre formation in CB-Obs (Figure 1d,e). Thus, while UDPG treatment of CB-Obs did not always result in discernable outcomes, the effects of P2Y_14_ inhibition were consistently seen, thereby supporting the presence of functional P2Y_14_ in primary murine osteoblasts. 

### 2.2. Generation of P2Y_14_ Knockout Cell-Line with CRISPR-Cas9

To further study the P2Y_14_ function in osteoblasts, we used BMP2-transfected C2C12 osteoblast-like cells (C2-Ob) that express P2Y14 (Figure 2a). Using CRISPR-Cas9, two P2Y_14_ knockout (Y14_KO_) clonal populations were independently generated and validated. In clonal Y14_KO_ cells, we observed ~10 bp deletion in P2Y_14_ gDNA (Figure 2b, left) which coincided with reduced protein expression (Figure 2b, *right*). Additionally, RT-qPCR melt-curve analysis showed that Y14_KO_ amplicons had a -4.9 °C shifted dissociation peak (Figure 2c) and P2Y_14_ mRNA expression was decreased by 87–99% (Figure 2d). These data confirm that P2Y_14_ expression was disrupted in both clonal knockouts which we then pooled for subsequent experiments. 

We next investigated P2Y_14_-mediated signalling in parental (wild type, Wt) and Y14_KO_ C2-Ob cells. Surprisingly, UDPG did not evoke [Ca^2+^]_i_ elevations in Fura2-loaded Wt cells. However, in Y14_KO_ cells [Ca^2+^]_i_ responses to UDPG were potentiated (*p* < 0.001, 2-way ANOVA), but had no discernable dose-dependency (Figure 2e). FK treatment of Wt cells—but not Y14_KO_ cells—resulted in cAMP elevation that was not reversed by UDPG (Figure 2f). Pharmacological inhibition of P2Y_14_ did not affect basal cAMP levels in Wt cells, while P2Y_14_ knockout resulted in significant reductions in basal cAMP levels (Figure 2f). Therefore, similar to primary osteoblasts, in C2-Ob cells, the inhibition of P2Y_14_ using pharmacological and genetic interventions had more distinct effects than the application of UDPG. 

### 2.3. P2Y_14_ Negatively Modulates Mechanical and Purinergic Signalling in Osteoblasts

The lack of discernable dose-dependency that accompanied the potentiated UDPG-induced [Ca^2+^]_i_ elevations in P2Y_14_-inhibited osteoblasts prompted us to investigate whether P2Y_14_ inhibition affects cellular sensitivity to mechanical stimulation that is known to induce purinergic signalling in bone cells [19]. We mechanically-stimulated osteoblasts by dispensing 10% media volume into Fura2-loaded osteoblast cultures (Figure 3a). This stimulus represented a relatively mild mechanical perturbation as indicated by a lack of response in untreated CB-Ob and Wt C2-Ob cultures. However, the same stimulus-induced pronounced [Ca^2+^]_i_ elevations in Y14_inh_-treated and Y14_KO_ cells (Figure 3b). Next, we mechanically-stimulated individual Fura2-loaded C2-Obs using a glass-micropipette, which we previously showed induces reversible cell membrane disruption [20], thus representing a more severe mechanical stimulus. The amplitude of mechanically-stimulated [Ca^2+^]_i_ elevations in Y14_KO_ C2-Ob cells was significantly higher than in Wt cells (Figure 3c,d). In addition, [Ca^2+^]_i_ responses in neighbouring non-stimulated C2-Ob cells, which are mediated by ATP and ADP released from the mechanically-stimulated cell [19,20,59], tended to have more oscillatory peaks (*p* = 0.09) with greater magnitudes (*p* = 0.09) (Figure 3e–g). When ATP and ADP were applied directly to Fura2-loaded Wt or Y14_KO_ cells, calcium responses to both agonists were significantly potentiated in Y14_KO_ cultures (Figure 3h). These data suggest that P2Y_14_ negatively modulates mechanical and purinergic signalling in osteoblastic cells, and when P2Y_14_ signalling is suppressed, cells become hypersensitive to stimulation. 

### 2.4. P2Y_14_ Signalling is Associated with Proliferation

To assess the functional outcomes of P2Y_14_ signalling, we first examined its role in cell proliferation. Cell counts were significantly reduced in Y14_KO_ compared to Wt C2-Ob cells (Figure 4a). Similarly, pharmacological inhibition in CB-Obs resulted in significantly reduced cell numbers on day 7 of culture, although the effect was of lower magnitude compared to Y14_KO_ (Figure 4b). UDPG treatment increased CB-Ob cell counts, which was reversed in the presence of Y14_inh_ (Figure 4b). As a measure of metabolic activity, we evaluated absorption of phenol red in conditioned media sampled from day 7 CB-Ob cultures and observed significant reductions in 560 nm absorption in UDPG-treated cultures, indicating increased extracellular acidity (Figure 4c). These effects were reversed when P2Y_14_ activity was inhibited (Figure 4c). We next examined if P2Y_14_ affected mitogenic or metabolic signalling in osteoblasts. Immunoblot analysis showed that 15 min UDPG treatment stimulated P2Y_14_-dependent ERK1/2 phosphorylation (pERK1/2) in C2-Ob (Figure 4 d,e). In CB-Obs UDPG did not induce any discernable changes in pERK1/2, however, inhibition of P2Y_14_ resulted in a tendency towards reduced pERK1/2 (*p* = 0.10). AMPK phosphorylation (pAMPK) was previously reported to suppress proliferation [60]. Stimulation with UDPG for 48 h leads to a significant reduction in pAMPKα levels in CB-Ob cultures, while the inhibition of P2Y_14_ had the opposite effect (Figure 4f,g). 

### 2.5. Role of P2Y_14_ in Osteogenic Differentiation

To determine the role of P2Y_14_ in osteogenic differentiation, primary bone-marrow cells were cultured with or without osteogenic factors and P2Y_14_ inhibitor, and osteoblast phenotype including osteogenic gene expression, alkaline phosphatase activity, collagen deposition and mineralization were evaluated at days 14 and 28 (Figure 5, Appendix A). In the presence of P2Y_14_ inhibitor, osteoblast gene expression on day 14 of differentiation was unaffected, except for COLA1 which was significantly higher (Figure 5a). Interestingly, this coincided with reduced collagen deposition (Figure 5c, Appendix A) and mineralization (Figure 5d, Appendix A) early in differentiation. At day 28 of differentiation, P2Y_14_ inhibition was associated with elevated expression of all osteogenic genes, including mature/late osteoblast/osteocyte markers DMP1 and SOST (Figure 5a). The earlier deficits in collagen deposition and mineralization had normalized by day 28 of differentiation (Figure 5c,d; Appendix A). These findings suggest that elevated gene expression observed in the later stages of differentiation of Y14_inh_-treated cultures serves as a compensatory mechanism for earlier differentiation delays. Importantly, these data demonstrate that P2Y_14_ is not necessary for osteogenic differentiation but can act as a modulator of osteogenic differentiation. 

### 2.6. Skeletal Phenotype in P2ry14^-/-^ Mice

To evaluate the skeletal phenotype of P2Y_14_ knockout mice, we used publicly available data from the International Mouse Phenotyping Consortium (IMPC). 9-week-old female *P2ry14^-/-^* mice weighed 1.22 g (95% CI: 0.06, 2.39) more than Wt controls, but this difference was not seen in their male counterparts (Figure 6a). Bone mineral content (BMC; Figure 6b) and density (BMD; Figure 6c) was unaffected in 13–14-week-old female *P2ry14^-/-^* mice. However, the tibial lengths of female and male *P2ry14^-/-^* mice were 0.51 mm (95% CI: 0.12, 0.91) and 0.85 mm (95% CI: 0.71, 0.99) longer than corresponding wild-type tibiae (Figure 6d). Thus, consistent with in vitro results, P2Y_14_ is not necessary for bone formation and maintenance in vivo, however tibial phenotype suggests that it may play a modulatory role in skeletal development. 

## 3. Discussion

### 3.1. Overview

We demonstrated the presence of functional P2Y_14_ receptor in murine osteoblasts using genetic and pharmacological approaches. While UDPG stimulation of P2Y_14_ did not always result in discernable outcomes, P2Y_14_ inhibition led to potentiated calcium signalling and altered basal cAMP in both osteoblast models. Osteoblasts exposed to mechanical or purinergic stimulation exhibited potentiated calcium responses when P2Y_14_ was inhibited, suggesting that P2Y_14_ negatively modulates cellular sensitivity to mechanotransductive stimuli. Functionally, inhibition of P2Y_14_ signalling reduced osteoblastic proliferation, likely through MAPK- and AMPK-related pathways. While overall osteoblastic differentiation was unimpeded in the absence of P2Y_14_, we have found that P2Y_14_ modulated the timing of osteogenic gene expression, collagen deposition and mineralization. The skeletal phenotype of *P2ry14^-/-^* mice corroborated in vitro data, suggesting that P2Y_14_ is not necessary for bone formation, but modulates skeletal development resulting in differences in tibial length. Taken together, these data imply that P2Y_14_ in osteoblasts plays a regulatory role by reducing cell responsiveness to mechanical stimulation and mechanotransductive signalling. 

### 3.2. P2Y_14_ Signal Transduction

We investigated the coupling of P2Y_14_ with [Ca^2+^]_i_ and cAMP-dependent signal transduction. In heterologous expression systems, P2Y_14_ was demonstrated to couple with recombinant Gα_16_ (Gα_q/11_ family) to induce calcium signalling [9,53], and with Gα_i/o_ to inhibit adenylate cyclase (AC) [9,53,61]. We have found that UDPG stimulation led to dose-dependent [Ca^2+^]_i_ elevations in CB-Obs, but not C2-Obs. The lack of UDPG-induced [Ca^2+^]_i_ response in P2Y_14_-expressing C2-Obs is similar to that reported in neutrophils, which also express P2Y_14_ but fail to evoke UDP-induced [Ca^2+^]_i_ responses [62]. The presence of oscillatory [Ca^2+^]_i_ elevations observed in CB-Obs is consistent with the involvement of store-operated [Ca^2+^]_i_ mobilization downstream of phospholipase C (PLC) activation, rather than Ca^2+^ influx through a membrane channel. This is consistent with previous findings that calcium signalling by endogenously-expressed P2Y_14_ is PLC-dependent in glioma C6 cells, immature monocyte-derived dendritic cells, and RBL-2H3 mast cells [63,64,65]. In both osteoblastic models, P2Y_14_ inhibition consistently resulted in potentiated calcium signalling.

We demonstrated that UDPG did not inhibit FK-induced cAMP accumulation in either osteoblast model, which was consistent with another study that reported similar findings in P2Y_14_-expressing HL-60 cells [66]. In contrast, UDPG was able to inhibit FK-induced cAMP in U373 MG astrocytoma [67], T-lymphocytes [68], and neutrophils [62], however, since P2Y_14_ was found to be absent in U373 MG cells [67], the effects of UDPG may be P2Y_14_-independent. We showed that inhibition of P2Y_14_ increased basal cAMP in CB-Ob cells, thereby demonstrating functional coupling of P2Y_14_ and Gα_i/o_; however, in C2-Ob *P2yr14^-/-^* cells cAMP levels were significantly reduced and unresponsive to FK treatment. The inconsistencies in cAMP responses observed between pharmacological and genetic P2Y_14_ perturbation experiments may relate to the duration of impaired P2Y_14_ function; From this perspective, the effects of acute pharmacological inhibition were consistent with expectations (i.e., function coupling of P2Y_14_ and Gα_i/o_), while effects of chronic impairment in *P2yr14^-/-^* cells were not, thus suggesting that over time cells may adapt to a novel state that exhibits distinct cAMP characteristics. Nonetheless, our data suggest that while UDPG induced some signalling events consistent with those previously described as P2Y_14_-mediated, its effects were variable in two osteoblastic cell lines and not always consistent with the outcomes of pharmacological or genetic inhibition of P2Y_14_. Together these data suggest that UDPG may not be a specific P2Y_14_ agonist in osteoblastic cells. 

To further investigate the potential origin of inconsistencies in P2Y_14_-induced signalling, we conducted a rapid review and meta-analysis of prior literature to quantitatively synthesize the dose-dependency of P2Y_14_ on its studied endogenous ligands (Appendix A, Figure 7). We found that in studies focused on the function of endogenously-express P2Y_14_, the potency of UDP-sugars was two orders of magnitude lower compared to the studies reporting results from heterologous expression systems (Figure 7a). When the data from more detailed experiments on exogenously expressed P2Y_14_ was examined (Figure 7b–d), no differences were observed between the responses of P2Y_14_ from human, mouse, or rat (Figure 7b), or when distinct UDP-sugars were applied (Figure 7c). However, EC_50_ for different signalling outcomes downstream of exogenously expressed P2Y_14_ varied by orders of magnitude, ranging from 12.6 nM (95% CI: 7.04, 22.5) for calcium signalling, to 81.9 nM (95% CI: 45.6, 147) for cAMP inhibition, to 341 nM (95% CI: 232 to 502) for IP_3_ accumulation. Importantly, the EC_50_ obtained for calcium signalling in the current study was consistent with our meta-analytic prediction of 8.9 µM (95% CI: 1.2 to 65.7) using pooled data from previous studies (Figure 7e, Appendix A). The difference between endogenously and exogenously expressed receptors may suggest that P2Y_14_ signalling is regulated by post-translational modifications. It was previously suggested that P2Y_14_-mediated calcium signalling is related to P2Y_14_ glycosylation since in glioma C6 cells only N-glycosylated P2Y_14_ was able to induce [Ca^2+^]_i_ elevations [63]. Moreover, it was shown that synthetic and microbial UDPG preparations induced different responses in N9 microglial cells: 15 µM synthetic UDPG inhibited phosphorylation of cAMP response element-binding protein (CREB), thereby indicating reduced cAMP levels, while 15 µM microbial UDPG had no effect [69]. This raises the concern that microbial UDPG preparations, which are used in most P2Y_14_ studies (including this study), contain contaminants that may mediate P2Y_14_-independent or mask P2Y_14_-dependent outcomes. Overall, we conclude that functional P2Y_14_ receptors are present in murine osteoblasts, however, UDP-glucose may not be the main P2Y_14_ ligand for osteoblasts.

### 3.3. Modulation of Mechanical and Purinergic Signalling 

Among the most exciting findings of this study was that P2Y_14_ was found to negatively modulate mechanical and purinergic signalling. The lack of discernable dose-dependency that accompanied potentiated UDPG-induced [Ca^2+^]_i_ responses in P2Y_14_-inhibited CB-Obs was contrary to expectations as another study had reported that P2Y_14_-knockdown in astrocytes abolished UDPG-induced [Ca^2+^]_i_ elevations [70]. However, upon observing similar potentiating effects in UDPG-insensitive C2-Ob cells, we realized that this potentiation was unlikely to be UDPG-dependent. We then found that a mild mechanical stimulation, that would normally go unperceived, was sufficient to induce robust [Ca^2+^]_i_ responses in cells with impaired P2Y_14_ signalling. This finding had three implications. First, the potentiated responses to UDPG observed in Y14_inh_-treated CB-Ob and Y14_KO_ C2-Ob were responses to mechanical stimulation and not to UDPG itself. Second, the emergent [Ca^2+^]_i_ elevations were not PPTN (Y14_inh_) off-target effects since they were reproduced in P2Y_14_-knockout C2-Obs. Third, PPTN likely inhibited UDPG-mediated [Ca^2+^]_i_ responses in CB-Ob, but this was masked by the potentiation of mechanically-induced responses. Consistent with this observation, we also found that inhibition of P2Y_14_ potentiated [Ca^2+^]_i_ responses to severe mechanical stimulation and ATP and ADP stimulation. It is possible that stronger responses to ATP and ADP were due to an additive contribution of unaffected P2R signalling and increased [Ca^2+^]_i_ responses to mechanical perturbations introduced during agonist application; however, disproportionate increases in cellular responsiveness to higher concentrations of both ligands suggest that potentiation of P2R signalling is likely.

Among the P2 receptor family, the P2Y_13_ receptor is another P2 receptor that has been implicated as a negative modulator of mechanically-induced osteogenesis [44,45]. P2Y_13_ ablation was suggested to enhance the mechano-adaptive response by reducing alkaline phosphatase expression, consequently resulting in elevated extracellular ATP levels. Although P2Y_14_ and P2Y_13_ are evolutionarily related [47,48], we found that P2Y_14_ inhibition did not affect alkaline phosphatase activity in differentiating osteoblasts. We also rejected the involvement of cAMP signalling in modulating mechano-sensitivity because Y14_inh_ treatment in CB-Obs and P2Y_14_-knockout in C2-Obs had opposing effects on basal cAMP levels. Other pathways affected by P2Y_14_ signalling may be involved in regulating mechanotransductive signalling. We showed that in CB-Ob cells, UDPG-induced actin reorganization was disrupted by P2Y_14_ inhibition, and consistent with this finding, the involvement of RhoA has been implicated downstream of P2Y_14_ [71]. Moreover, we demonstrated that UDPG and P2Y_14_ inhibition had opposing effects on phosphorylation of ERK1/2 and AMPKα, of which only the former has been previously demonstrated [62,72]. While these effects are not immediate, in P2Y_14_ inhibition studies there is sufficient time for cell transition to a new steady-state, which may be characterized by higher responsiveness to mechanotransductive signalling. 

### 3.4. Skeletal Phenotype

In our study, P2Y_14_ was not necessary for the differentiation of bone-forming osteoblasts. Antagonizing P2Y_14_ reduced collagen deposition and matrix mineralization at differentiation day 14, but it was normalized by day 28 suggesting that P2Y_14_ may influence collagen synthesis and deposition. In bone-resorbing osteoclasts, P2Y_14_ signalling has been reported to promote osteoclast formation [57,73]. P2Y_14_ knockdown in bone-marrow-derived monocytes significantly impaired RANKL-induced osteoclastogenesis [57], while UDPG application promoted osteoclast formation in vitro [57,73]. Evidence from in vitro osteoblast and osteoclast cultures suggests that in vivo P2Y_14_ deficiency would result in an osteopetrotic phenotype predominantly driven by impaired osteoclast formation. While there were no discernable differences in bone mineral density or content were observed in *P2ry14^-/-^* mice, this does not dismiss the possibility that skeletal microarchitecture (i.e., trabecular number, separation, thickness, etc.) is affected in these mice, and future histomorphometry and µCT analysis is needed to validate these findings. It is also possible that bone-embedded osteocytes which express P2Y_14_ but remain uncharacterized provide additional P2Y_14_-mediated contributions [54]. Alternatively, other P2Y_14_-dependent functions, particularly in immunity and inflammation [14] may mask the predicted phenotype in *P2ry14^-/-^* mice. 

### 3.5. Study Limitations and Future Directions

This study has several noteworthy limitations. *First*, we characterized the role of P2Y_14_ through pharmacological inhibition and genetic knockout and through these complementary approaches we identified certain inconsistencies. Whether this relates to varying degrees of intervention specificity or differences in acute versus chronic P2Y_14_ impairment is unclear. Overexpression studies—which to date are lacking—would help clarify outstanding questions (e.g., does P2Y_14_ overexpression attenuate cellular responsiveness to mechanical stimulation?) and provide insights into the complete spectrum of cellular effects that are mediated by P2Y_14_ receptor. Secondly, we have implicated P2Y_14_ in the modulation of mechanotransduction signalling, however, our study was limited to in vitro experiments, and we did not elucidate the mechanism by which this is achieved or how this influences the mechano-adaptive skeletal response in vivo. It is known that the cytoskeleton is intimately linked to mechano-sensitivity [74,75,76], and further work examining the link between P2Y_14_ modulation and the cytoskeleton may be warranted. Additionally, it will be necessary to evaluate the skeletal response to controlled mechanical loading in Wt and *P2ry14^-/-^* mice. Thirdly, we have demonstrated that P2Y_14_ is functionally expressed in osteoblasts through inhibition and knockout, but the response to UDPG varies between osteoblast models and does not evoke a complete response in certain pathways that are known to be functionally coupled to P2Y_14_. Further work is needed to determine whether UDPG is released by osteoblasts. Additionally, it will be pertinent to determine whether other UDP-sugars, such as UDP-galactose, are active agonists for P2Y_14_ in bone cells. 

### 3.6. Concluding Remarks

In this study, we established that functional P2Y_14_ is expressed and coupled to AC inhibition and [Ca^2+^]_i_ mobilization in primary murine osteoblasts. Using two osteoblast models, we demonstrated the effects of P2Y_14_ inhibition are consistently more pronounced and discernable than those induced by UDPG treatment, indicating that certain P2Y_14_-mediated effects, including AC inhibition (C2-Ob and CB-Ob), may not be inducible by endogenous ligand stimulation. Importantly, we showed for the first time that P2Y_14_-inhibition results in hypersensitivity to mechanical and purinergic stimulation, reflecting a possible modulatory role for P2Y_14_ in mechanotransductive purinergic signalling. Functionally, P2Y_14_ signalling regulated osteoblast proliferation and the timing of collagen synthesis and deposition during differentiation. P2Y_14_-deficient mice exhibited a mild skeletal phenotype with elongated tibiae and no differences in bone mineral density. Our findings demonstrate a modulatory role for P2Y_14_ in osteoblasts and furthers our understanding of how the P2 receptor network integrates mechanical and purinergic signals in the bone. 

## 4. Materials and Methods

### 4.1. Software. 

Figure preparation: CorelDRAW X8 (Corel, Ottawa, ON, Canada); Image analysis: ImageJ 1.52h (NIH, Bethesda, MD, USA); statistical analysis: MATLAB R2018a (MathWorks, Natick, MA, USA).

### 4.2. Reagents and Solutions. 

Reagent sources and solution recipes are provided in Appendix A Solution and Reagents section. 

### 4.3. Cell Culture. 

All procedures were approved by McGill’s University’s Animal Care Committee (protocol # 2012– 7127 approved 01 July 2018) and complied with the ethical guidelines of the Canadian Council on Animal Care. Bone marrow- cells and compact-bone-derived osteoblasts (CB-Ob cells) were isolated from 4–6 weeks old C57BL/6 (Charles River) as previously described [20]. CB-Obs were plated at 10^4^ cells/cm^2^ in osteoblast differentiation medium (αMEM supplemented with 10% FBS, 1% sodium pyruvate, 1% penicillin-streptomycin and 50 µg/mL ascorbic acid [AA]) and cultured for 2–3 days prior to experiments. The C2C12 cell line (ATCC CRL-1772) stably transfected with BMP-2 (C2-Ob cells) was plated at 10^4^ cells/cm^2^ in DMEM (supplemented with 10% FBS, 1% sodium pyruvate, 1% penicillin-streptomycin) and cultured for 2–3 days before experiments. Bone marrow cells were plated at a density of 5 × 10^4^ cells per cm^2^ in osteoblastic differentiation medium supplemented with 2 mM β-glycerol phosphate (βGP) to promote mineralization and media was refreshed every 2–3 days. The osteoblast phenotype was assessed at days 14 and 28. The absence of mycoplasma contamination was verified in cryo-preserved stocks of C2-OB cells using a PCR-based detection kit. 

### 4.4. Generation of P2Y_14_ Knockout Cell Line. 

C2-Ob cells were plated in 6-well plates at 100,000 cell/well density 2 days before transfection. On the day of transfection, 7.5 µL lipofectamine was diluted in 125 µL Opti-MEM medium (Solution A) and 5 µg *P2ry14* CRISPR/Cas9 plasmid and 10 µL P3000 reagent were diluted in a separate 125 µL aliquot of Opti-MEM (Solution B). Solutions A and B were then pooled in a 1:1 ratio and incubated at room temperature for 15 min. Cell media was aspirated and 250 µL of the pooled DNA-lipid complex solution was added to cells and left to incubate at 37 °C for 3 days. 3 days post-transfection, cells were visualized using a fluorescent microscope to verify successful transfection through the presence of GFP-positive cells. Transfected cultures were transferred to fresh DMEM media and treated with 5 µM puromycin for 7 days to select for puromycin-resistant clones. After selection, cells were transferred into puromycin-free media, allowed 3 days for recovery, and re-plated in 96 well plates at a ~1 cell per well density. After 3 weeks of expansion, half of each single-cell colony was re-plated in 96-well plates and the other half was collected for genomic DNA extraction using the DNeasy kit. Genomic DNA for each single-cell colony was amplified by touchdown PCR using primer sets designed to flank the genomic region targeted by Cas9 (Appendix A), and amplicons were separated on a gel to screen for clones with evident band shifts. Selected clones were subsequently validated by qRT-PCR and immunoblot analysis, and termed Y14_KO_ cells. Note that the *P2ry14* CRISPR/Cas9 plasmid that we used encoded a double-nickase variant of Cas9 which enhances the specificity of Cas9 and reduces the risk of off-target effects [77].

### 4.5. Pharmacological Inhibition of P2Y_14_

In specified experiments, P2Y14 was inhibited using the 4,7-disubstituted 2-naphthoic acid derivative, PPTN (termed Y14_inh_ in this study), which is potent and selective towards P2Y_14_ [78]. 100 nM Y14_inh_ dosing (0.1% DMSO as a vehicle) was informed by prior work demonstrating that Y14_inh_ has a nanomolar IC_50_ [49]. For longer-term cultures (e.g., osteoblast differentiation), media was refreshed every 2–3 days with newly added Y14_inh_. 

### 4.6. Quantitative Real-Time Polymerase Chain Reaction (qRT-PCR)

Total RNA was isolated using the RNeasy kit and QIAshredder columns and reverse transcribed using cDNA reverse transcription kit. Real-time qPCR was performed using the QuantStudio 7 Flex PCR System, with SYBR Green or TaqMan Master Mix. Primers and TaqMan probes are specified in Appendix A and cycling conditions in Appendix A. The dissociation characteristics of *P2ry14* amplicons were assessed for Wt and Y14_KO_ C2-Ob samples through melt-curve analysis [79]. 

### 4.7. Fluorescent Microscopy

For immunofluorescence, formalin-fixed cultures (pH 7.4, 8 min) were blocked in 5% BSA/PBST (1 h) and incubated overnight (4 °C) with P2Y_14_ antibody (1:1000 dilution). Secondary anti-rabbit FITC-conjugated secondary antibody was applied for 1 h (room temperature). For actin visualization, fixed cultures were stained with 150 nM phalloidin (15 min, room temperature). Nuclei were counterstained with DAPI (5 min, rt). Stained cultures were visualized with an inverted fluorescent microscope (Nikon T2000, Tokyo, Japan) and stress fibre-positive cells were visually classified as those in which long fibres were observed along the major cellular axis [80]. 

### 4.8. Intracellular Calcium Recordings and Analysis

Cells were plated on glass-bottom 35 mm dishes or 48-well plates (MatTek Corporation), for single-cell mechanical stimulation and agonist application experiments, respectively. Cells were loaded with Fura2-AM for 30 min, acclimatized in physiological solution (PS) for 10 min on the stage of an inverted fluorescence microscope (Nikon T2000), and imaged as described previously [20]. The calcium response parameters were analyzed using a previously developed MATLAB algorithm [81]. To assess UDPG, ATP, and ADP dose-dependencies, Fura2-loaded C2-Ob or CB-Ob cells were bathed in 270 µL PS and 30 µL of UDPG, ATP or ADP solutions at 10× final concentration were added (e.g., 30 µL of 10 µM ATP solution was added to cells to achieve 1 µM ATP stimulation). 

### 4.9. Intracellular cAMP Assay

C2-Ob and CB-Ob cells plated 2 days before the experiment (5000 cells/well in 96-well plate) were treated as specified and 30 min later media was aspired, and the plate was frozen (−80 °C). Sample preparation and intracellular cyclic AMP measurement were performed using cAMP Select ELISA Kit (Cayman Chemical, Ann Arbor, MI, USA).

### 4.10. Immunoblotting

Cell lysates were extracted in RIPA lysis buffer and samples were prepared and subject to SDS-PAGE on a 10% (w/v) acrylamide gel as previously described [20]. Blotted nitrocellulose membranes were incubated with primary antibodies overnight (1:1000 dilution, 5% BSA in TBST, 4 °C) and secondary antibodies were applied for 1 h (1:1000 dilution, 5% BSA in TBST, rt) before visualization with chemiluminescence system. 

### 4.11. Mechanical-Stimulation

#### 4.11.1. Mild Mechanical Stimulus

10% media volume was dispensed into cell culture (e.g., 30 µL physiological solution added to cells bathed in 270 µL physiological solution). This is a milder approach than that taken by Kowal and colleagues who showed that pump injection of 33% media volume into cell cultures can evoke mechanical responses [82]. 

#### 4.11.2. Severe Mechanical Stimulus

Single osteoblastic cells were stimulated by local membrane indentation with a glass micropipette using a FemtoJet microinjector NI2 (Eppdendorf Inc., Hamburg, Germany), as previously described [20]. 

### 4.12. Proliferation Assays

Cells were plated (5000 cells/well in 48 well plate) and cultured for 5–7 days. For nuclear count assessment, cultures were fixed with formalin (pH 7.4, 8 min) at specified time points, stained with DAPI, and washed with H_2_O. Representative images of each well were taken at 4× magnification and the ‘find maxima’ function in ImageJ was used to count the number of nuclei in each image. To monitor extracellular pH (known to change as a function of cellular lactate and CO_2_ production [83]), phenol red absorption at 560 nm was measured in live cell cultures. 

### 4.13. Osteoblast Differentiation Assays

#### 4.13.1. Alkaline Phosphatase (ALP) Staining

ALP staining solution was prepared fresh and formalin-fixed cultures were stained as previously described [20]. ALP-positive cells were stained pinkish-red.

#### 4.13.2. Collagen Deposit and Mineralized Nodule Staining

Osteoblast cultures were fixed in 70% EtOH (1 h, 4 °C) and collagen deposits were stained using Picro-Sirius Red stain kit (according to manufacturer’s protocol), or calcium deposits were visualized using 1% Alizarin red solution (pH 4.5, 5 min, room temperature). Staining for ALP, collagen or calcium deposits was visualized using bright-field microscopy and surface area coverage (%) was quantified in ImageJ [84].

### 4.14. In vivo Knockout Phenotype

Phenotypic data for P2Y_14_ knockout mice were obtained from the International Mouse Phenotyping Consortium (IMPC) that curates a publicly accessible database of mutant mice generated and phenotyped in designated IMPC phenotyping centres (http://www.mousephenotype.org). The production and phenotypic analysis of P2Y_14_^em1(IMPC)H^ mutant mice (MGI ID: 5749947; C57BL/7NTac genetic background) was performed at the MRC Harwell Center (Oxfordshire, UK) [85]. Bodyweight (Identifier: IMPC_DXA_001_001), Bone mineral density (BMD; IMPC_DXA_004_001) and content (BMC; IMPC_DXA_005_001) were assessed using DEXA (dual-energy X-ray absorptiometry). Tibial length (IMPC_XRY_033_001) was determined from digital X-ray images obtained from immobilized mice using a Faxitron X-Ray system or NTM digital X-ray scanner. 

### 4.15. Rapid Review and Meta-Analysis

Rapid literature reviews were performed to (*i*) evaluate P2Y_14_ expression in bone-residing cells and (*ii*) determine P2Y_14_ dose-response to endogenous ligands. The search strategy and inclusion criteria for both reviews are provided in Appendix A and study-level characteristics in Table 1 and Appendix A, respectively. In the latter review, dose-dependency data were extracted from each study using the ‘data extraction’ MetaLab module [19] and data were fit to hill functions to estimate the agonist concentration that induces half-maximal response (EC_50_):(1)$$Y=\frac{y_{max}X^{\beta }}{EC_{50}^{\beta }+X^{\beta }}$$
where X are agonist concentrations, Y are measured responses, $y_{max}$ is the maximal response parameter and β is a slope-related parameter. Study-level EC_50_ estimates were pooled using a random-effects model and interstudy variance was estimated by DerSimonian Laird estimator [19]. The percentage of heterogeneity (i.e., inconsistency between pooled datasets) explained by subgroup analysis was computed as $R_{\exp}^{2}$ [19]. EC_50_ values for UDPG-induced [Ca^2+^]_i_ response mediated by endogenous murine P2Y_14_ were predicted by a meta-regression model fit using uridine agonists (UDP, UDP-glucose, UDP-galactose, UDP glucuronic acid), P2Y_14_ source (endogenous, exogenous), P2Y_14_ species (human, mouse, rat) and the type of measured response (cAMP, [Ca^2+^]_i_) as predictor variables. 

### 4.16. Statistical Analysis

Data are presented as representative images, means ± standard error (sem) or means ± 95% confidence intervals (95% CI), as specified in each figure panel along with sample sizes *N* (number of independent experiments; biological replicates) and *n* (number of technical replicates). When multiple technical replicates *n* were obtained for a given experiment *N*, the means of technical replicates were treated as single samples, and the statistical inference was performed at the level of the biological replicates. Curve fitting and [Ca^2+^]_i_ transient characterization were performed in MATLAB. Statistical significance was assessed by one- or two-way ANOVA (as specified) and post-hoc two-way unpaired Students t-tests were adjusted using the Bonferroni correction. Significance levels were reported as single symbol (* *p* < 0.05), double symbol (** *p* < 0.01), or triple symbol (*** *p* < 0.001).

## Figures and Tables

**Figure 1 ijms-21-02747-f001:**
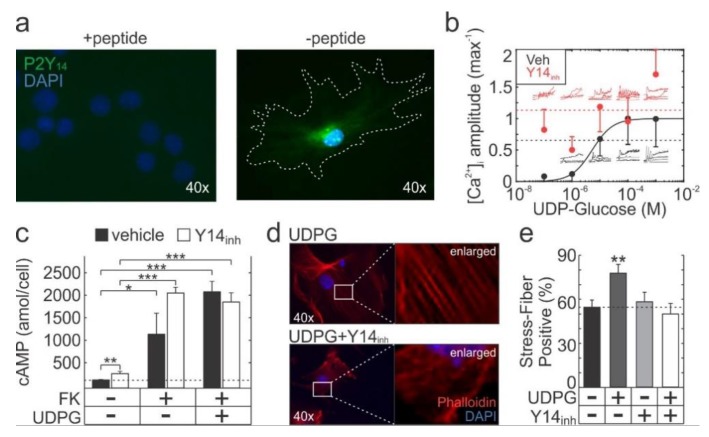
P2Y_14_ expression and signalling in primary osteoblasts. (**a**) Representative image of P2Y_14_ immunofluorescence in CB-Ob cells. Formalin-fixed cells were incubated with P2Y_14_ antibody (with or without blocking peptide) followed by secondary FITC-conjugated antibody and nuclei were counter-stained with DAPI. (**b**) Fura2-loaded CB-Ob cells pretreated 10 min with 0.1% DMSO (vehicle; veh) or 100 nM PPTN (Y14_inh_) were stimulated with UDP-glucose (UDPG) and [Ca^2+^]_i_ elevation amplitudes were estimated from live-cell recordings. Solid black line: Hill function, dashed lines: Global means (pooled across all concentrations). Data are means ± sem, *n* = 3 independent experiments with 5–12 cells imaged per experiment. (**c**) Intracellular cAMP concentrations in CB-Ob cultures pretreated with vehicle or Y14_inh_ (10 min) and stimulated with 10 µM forskolin (FK) and/or 10 µM UDPG (30 min) before collecting samples for measurement. Data are means ± sem, *n* = 3 independent cell cultures. (**d**,**e**) Actin cytoskeleton in phalloidin-stained CB-Ob cultures pretreated with vehicle or Y14_inh_ (10 min) and stimulated with UDPG (10 min). Stress fibre prevalence was determined as the percentage of cells that exhibited stress-fibres. Representative images for UDPG and UDPG+Y14 conditions, along with contrast-enhanced enlargements are shown (**d**). Data are means ± sem, *n* = 2 independent cultures, 13–18 cells assessed per culture per condition (**e**). Significance assessed by t-test with Bonferroni correction; *p* < 0.05 *, *p* < 0.01 ** and *p* < 0.001 ***.

**Figure 2 ijms-21-02747-f002:**
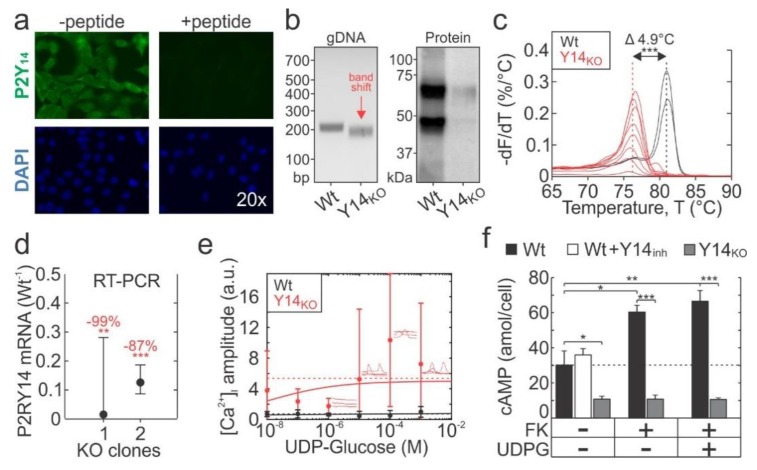
Generation of P2Y_14_ knockout C2-Ob cell line with CRISPR-Cas9. (**a**) P2Y_14_ immunofluorescence in C2-Ob cells. Fixed cells were incubated with a P2Y_14_ antibody (with or without blocking peptide) and then with secondary FITC-conjugated antibody and counter-stained with nuclear DAPI stain. (**b**) P2Y_14_ disruption by CRISPR-Cas9 assessed at the level of genomic DNA- (gDNA, by PCR; left panel) and protein (by immunoblotting; right panel). (**c**) Wild-type (Wt) and P2Y_14_ knockout (Y14_KO_) amplicon melting analysis performed by RT-PCR. (**d**) P2Y_14_ gene expression in two independently isolated Y14_KO_ clones, relative to Wt cells. Data are means ± 95% CI, *n* = 2 independent experiments, 2–4 technical replicates per experiment. (**e**) Fura2-loaded Wt and Y14_KO_ cells were stimulated with UDPG and [Ca^2+^]_i_ elevation amplitudes were estimated from live-cell recordings. Solid curves indicate the Hill function; dashed lines show the global means. Data are means ± sem, *n* = 3 independent cultures, 8–19 cells recorded per culture. (**f**) Intracellular cAMP concentrations in Wt and Y14_KO_ cells stimulated with 10 µM forskolin (FK) and/or 10 µM UDPG for 30 min before collecting samples for measurement. Data are means ± sem, *n* = 3 independent cultures. Significance assessed by t-test with Bonferroni correction; *p* < 0.05 *, *p* < 0.01 ** and *p* < 0.001 ***.

**Figure 3 ijms-21-02747-f003:**
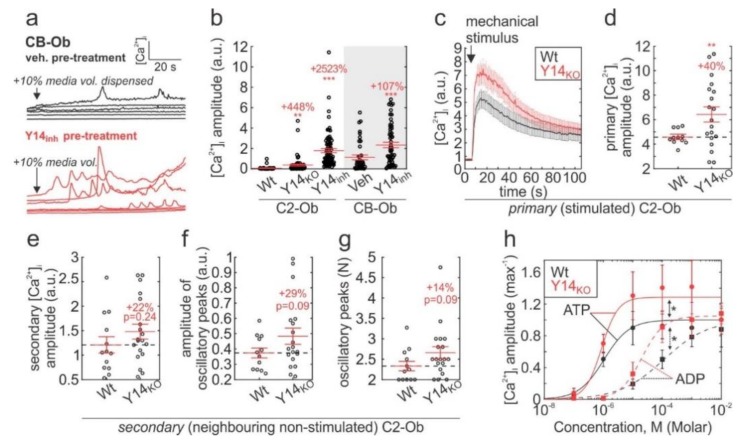
P2Y_14_ involvement in tuning mechanical- and purinergic-signalling in osteoblasts. (**a**,**b**) Fura2-loaded CB-Ob and C2-Ob (Wt or Y14_KO_) cells were pretreated with vehicle (0.1% DMSO) or 100 nM PPTN (Y14_inh_) for 10 min, and then mechanically-stimulated by dispensing 10% media volume into culture; (**a**) Representative [Ca^2+^]_i_ recordings in stimulated CB-Ob; (**b**) [Ca^2+^]_i_ elevation amplitudes. Raw data and means ± sem are shown, *n* = 4–5 independent cultures, 9–20 cells recorded per culture. (**c**–**g**) Individual Fura2-loaded Wt or Y14_KO_ C2-Obs were mechanically-stimulated by micropipette. (**c**) [Ca^2+^]_i_ responses and (**d**) amplitudes of [Ca^2+^]_i_ responses in the primary (stimulated) cell. Data are means ± sem, *n* = 14–20 independently stimulated cells (i.e., trials). (**e**–**g**) [Ca^2+^]_i_ responses in secondary (neighbouring non-stimulated) cells were analyzed for their amplitude (**e**), and amplitude (**f**) and number (**g**) of oscillatory peaks. Raw data and means ± sem are shown, *n* = 14–20 trials with 8–23 cells recorded per trial. (**h**) Fura2-loaded Wt and Y14_KO_ C2-Obs were stimulated with ATP or ADP, and [Ca^2+^]_i_ elevation amplitudes were determined. Solid and dashed curves are the Hill functions that fit the ATP and ADP data, respectively. Data are means ± sem, *n* = 3 independent cultures, 8–23 cells recorded per culture. For (**b**–**g)**, ** *p* < 0.01 and *** *p* < 0.001 significance assessed by ANOVA and post-hoc Bonferroni test (versus Wt or Veh conditions); for h, * *p* < 0.05 significance determined by 2-way ANOVA.

**Figure 4 ijms-21-02747-f004:**
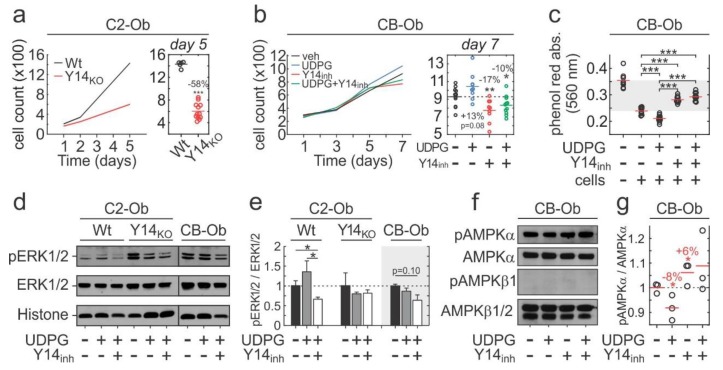
P2Y_14_ regulates osteoblast proliferation. (**a,b**) C2-Ob (**a**) and CB-Ob (**b**) cells were plated at equal densities, cultured for the indicated number of days, formalin-fixed, and DAPI-stained. The number of cells/field was counted at specified days. Raw data (day 5 for C2-Ob, **a**—right panel; day 7 for CB-Ob, **b**—right panel) and means are shown, *n* = 5–12 independent cultures. (**c**) Absorption (560 nm) of phenol red in conditioned media sampled from CB-Ob cultures at day 7 was evaluated by spectrophotometry. Shaded gray region: Region between positive (+ cells) and negative (- cells) control. Raw data and means are shown, *n* = 9–11 independent cultures. **(d,e)** Immunoblot (**d**) and analysis (**e**) of phosphorylated and total ERK1/2 in C2-Obs and CB-Obs pretreated with (+) or without (−) 100 nM PPTN (Y14_inh_, 10 min) and stimulated with 10 µM UDPG (15 min). Data are means ± sem, *n* = 3 independent cultures. (**f,g**) Immunoblot (**f**) and analysis (**g**) of AMPKα and AMPKβ1/2 phosphorylation in CB-Obs after 48 h pre-treatment with or without Y14_inh_ and/or 100 µM UDPG. Raw data and means ± sem are shown; *n* = 3 independent cultures. Significance assessed by t-test with Bonferroni correction; *p* < 0.05 * and *p* < 0.001 ***.

**Figure 5 ijms-21-02747-f005:**
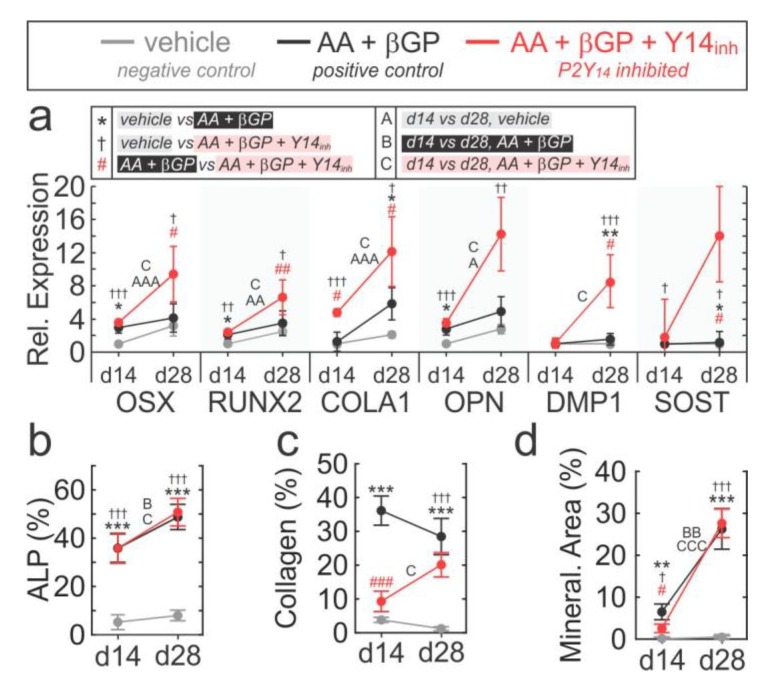
Role of P2Y_14_ in osteogenic differentiation. Bone marrow-derived cells were cultured with or without osteogenic factors (50 µg/mL ascorbic acid, AA; 2 mM β-glycerol phosphate, βGP) and 100 nM PPTN (Y14_inh_) and osteogenic phenotype was evaluated at days 14 and 28. (**a**) *Top*: description of symbols representing specific comparisons; the level of significance is indicated by single (*p* < 0.05), double (*p* < 0.01) or triple (*p* < 0.001) symbols. *Bottom*: gene expression of osteogenic markers assessed by RT-PCR at days 14 and 28 post-osteogenic induction. Data means ± sem, normalized to d14 vehicle condition; *n* = 5 independent cultures. (**b**,**c**) Cultures were fixed and stained for alkaline phosphatase (**b**; ALP), collagen (**c**; Sirius red stain) and mineralization (**d**; alizarin red stain). Stained cultures were visualized by brightfield microscopy, and surface coverage (%) was quantified; *n* = 5 independent cultures, with 2–5 fields of view quantified per culture.

**Figure 6 ijms-21-02747-f006:**
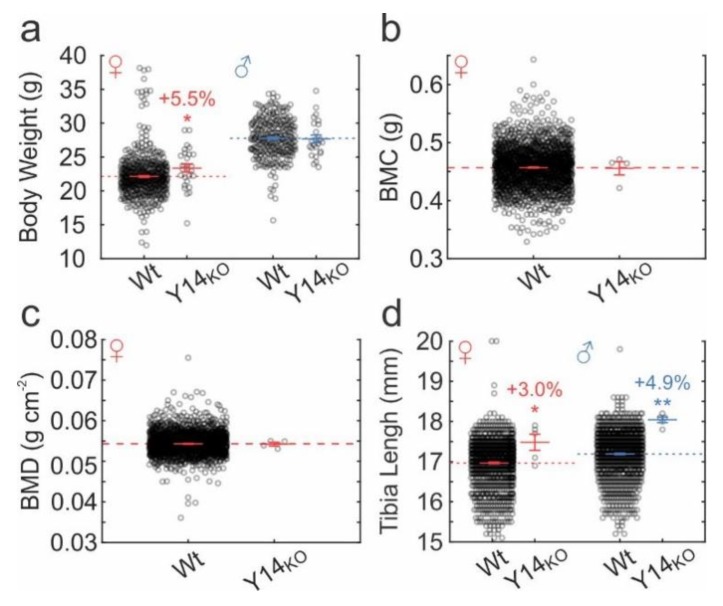
Skeletal phenotype of P2Y_14_ mutant mice. Wt and Y14_KO_ mouse phenotype data were obtained from the IMPC database. (**a**) body weight (9 week-old mice). *n* = 288–556 Wt, 24–27 KO mice/sex. (**b**) Bone mineral content, BMC (13–14 week-old mice). *n* = 1814 Wt, 4 KO mice. (**c**) Bone mineral density, BMD (13–14 week-old mice). *n* = 1807 Wt, 4 KO mice. (**d**) Tibial lengths (13–14 week-old mice). *n* = 1061–1093 Wt, 5 KO mice/sex. Raw data and means ± sem are shown. Significance was assessed by pairwise t-test; * *p* < 0.05 or ** *p* < 0.01.

**Figure 7 ijms-21-02747-f007:**
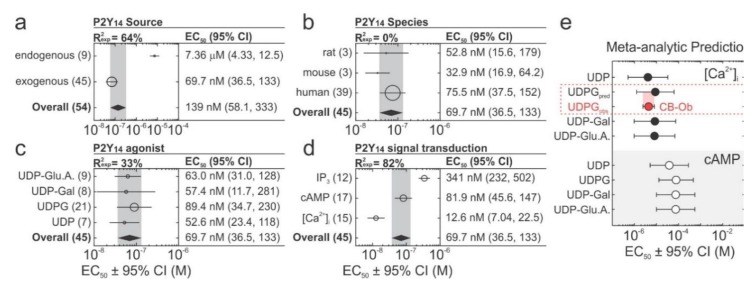
A meta-analysis of P2Y_14_ signalling. (**a**–**d**) Study-level P2Y_14_ EC_50_ values were estimated by hill functions (fitted to study-level data) and pooled using a random-effects model to obtain subgroup EC_50_ estimates for (**a**) P2Y_14_ source (endogenous or exogenous); (**b**) P2Y_14_ species (human, mouse, or rat); (**c**) uridine agonists used to evoke the response (UDP, UDP-glucose, UDP-galactose, or UDP glucuronic acid); and (**d**) measured outcome (IP_3_, cAMP, or [Ca^2+^]_i_). Panels **b**–**d** only included exogenously expressed P2Y_14_ data, and IP_3_ and [Ca^2+^]_i_ responses were measured in heterologous expression systems using chimeric Gα protein. $R_{exp}^{2}$: Percentage of heterogeneity (i.e., inconsistency between datasets) explained by specified covariates, *circles/lines*: subgroup means ± 95% CI (marker sizes are proportional to the number of datasets that are shown in parentheses), *diamonds/grey bands*: overall means ± 95% CI. (**e**) The meta-regression model was used to predict EC_50_ of [Ca^2+^]_i_ response induced by endogenously expressed murine P2Y_14_, and prediction (UDPG_pred_) was compared to observed data in CB-Obs (UDPG_obs_). UDP-Glu.A, UDP-glucuronic acid; UDP-Gal, UDP-galactose; UDPG, UDP-glucose.

**Table 1 ijms-21-02747-t001:** P2Y_14_ expression in bone. A rapid review of literature assessing the expression (+) or absence (−) of P2Y_14_ in bone-residing cells. The search strategy is provided in Appendix A. FC: flow cytometry, IF: Immunofluorescence, RT-PCR: reverse transcription-polymerase chain reaction, WB: Western blot.

Study	Sample	Species	Method	+/-
**Mesenchymal stem cells (MSC)**				
Ali 2018 [50]	Adipose-derived MSC	Human	RT-PCR	+
Kotova 2018 [51]	Adipose-derived MSC	Human	RT-PCR	+
Zippel 2012 [52]	Adipose-derived MSC	Human	RT-PCR	+
	Dental-follicle MSC-like cells	Human	RT-PCR	+
**Osteoblasts**				
Moore 2003 [53]	HOS osteosarcoma cell line	Human	RT-PCR	+
	MG63 osteosarcoma cell line	Human	RT-PCR	-
	SAOS2 osteosarcoma cell line	Human	RT-PCR	-
Paic 2009 [54]	Primary calvarial osteoblasts	Mouse (CD-1)	RT-PCR	+
Orriss 2012 [55]	Primary calvarial osteoblasts	Rat (Sprague-Dawley)	RT-PCR	+
Zippel 2012 [52]	Osteoblasts differentiated from adipose-derived MSC	Human	RT-PCR	+
	Osteoblasts differentiated from dental follicle cells	Human	RT-PCR	+
Mikolajewicz 2020 (*current study*)	C2C12-BMP2 osteoblast cell line	Mouse (C3H)	IF, RT-PCR, WB	+
	Primary compact bone-derived osteoblasts	Mouse (C57BL/6)	IF	+
**Osteocytes**				
Chambers 2000 [9]	Bone tissue	Human	RT-PCR	-
Paic 2009 [54]	Primary calvarial osteocytes	Mouse (CD-1)	RT-PCR	+
**Hematopoietic bone-marrow cells**				
Chambers 2000 [9]	Bone marrow cells	Human	RT-PCR	+
Cho 2014 [56]	Bone-marrow-derived hematopoietic stem cells	Human	RT-PCR	+
			FC	+
Lee 2013 [57]	Bone-marrow derived monocytes/macrophages	Mouse (C57BL/6)	RT-PCR	+
			WB	+
Lee 2003 [58]	Fetal bone marrow cells	Human	RT-PCR	+
			FC	+
**Osteoclasts**				
Lee 2013 [57]	Primary osteoclasts	Mouse (C57BL/6)	RT-PCR	+
			WB	+

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
