# Peer review of "Role of UDP-Sugar Receptor P2Y14 in Murine Osteoblasts"

_ijms, 2020, doi:10.3390/ijms21082747_

Round 1

Reviewer 1 Report

This is a well-written manuscript. The authors preliminary demonstrated the functionality of P2Y14 in a modulatory role in mechanotransductive purinergic signalling. The experimental design and the result discussion have well supported the hypotheses of the authors. All the figures and tables were nicely presented and organized. The reviewer has only one small suggestion to the authors, which is that the authors may add more discussion on the modulatory role for P2Y14 in osteoblasts on how the P2 receptor network integrates mechanical and purinergic signals in bone. The authors may be able to compare the recent advances and updates from the current literature with the results from this study.

Author Response

Response to reviewer 1

This is a well-written manuscript. The authors preliminary demonstrated the functionality of P2Y14 in a modulatory role in mechanotransductive purinergic signalling. The experimental design and the result discussion have well supported the hypotheses of the authors. All the figures and tables were nicely presented and organized. The reviewer has only one small suggestion to the authors, which is that the authors may add more discussion on the modulatory role for P2Y14 in osteoblasts on how the P2 receptor network integrates mechanical and purinergic signals in bone. The authors may be able to compare the recent advances and updates from the current literature with the results from this study.

We thank the reviewer for their favorable evaluation of our manuscript.

In response to the reviewer’s comments, we performed an updated search of the literature (April 2nd, 2020; initial search performed Sept 7th, 2018) using the search strategy designed to identify studies reporting P2RY14 expression in bone-residing tissue (Table S1, review 1). We identified 5 additional articles, of which 1 reported P2RY14 expression (by RT-PCR) in human adipose-derived MSC cells. Table 1 and Table S1 have now been revised to include this addition study (Ali et al. (2018) Purinergic Signal. 14(4):371-384). Similarly, we performed an updated search of the literature (April 2nd, 2020; initial search on July 13th, 2018) using the search strategy designed to identify studies reporting P2Y14 responses to endogenous ligands (Table S1, review 2). In this search we identified an additional 5 articles of which 1 was eligible for inclusion in Table S2 (summary of P2Y14 induced responses), but none were eligible for our meta-analysis which focused on cAMP, Ca2+ and IP3 mediated signaling. Nonetheless, the additional study that was identified was relevant to our discussion and was cited appropriately (Lin et al. (2019). J Cell Physiol. 234(11): 21199-21210).

In addition to the updated literature search, we have now provided additional discussion relating to the modulatory role of P2Y14 and have emphasized the limitations of our study along with future experiments that are warranted to further elucidate the modulatory role of P2Y14 in bone.

Reviewer 2 Report

This submission used in vitro studies to suggest that P2Y14 in osteoblasts reduces cell sensitivity to mechanical stimulation and modulates osteoblast differentiation. I like to give the following comments.

  1. P2X receptors and UDP-sugars must introduce in clear, particularly the functional role.
  2. In bone, role of P2 receptor needs in vivo evidence such as the association with exercise and others.
  3. Merit(s) of this study did not introduce in detail. Additionally, source of P2ry14 CRISPR/Cas9 plasmid and pharmacological inhibitors remained unknown.
  4. Mild mechanical stimulus needs the reference(s). Pharmacological inhibition of P2Y14 did not show the same effect on cAMP as that in receptor knock-out cell-line. How to support the specificity of inhibitor?
  5. P2Y14 seems involved in skeletal development but not for bone formation and maintenance. What is the role in functions?
  6. Role of P2Y14 receptor in murine osteoblasts has been characterized using the deletion and/or inhibition. What is the influence of overexpression?
  7. It has been demonstrated that UDPG and P2Y14 inhibition had opposing effects on phosphorylation of ERK1/2 and AMPK. But UDPG may not be the main P2Y14 ligand for osteoblasts. How to explain this truth?
  8. P2Y14-inhibition results in higher sensitivity to mechanical and purinergic stimulation. This novel view lacks the mechanism(s).
  9. Limitation(s) of this report may assist the unclear point(s) from obtained results.

Author Response

Response to reviewer 2

This submission used in vitro studies to suggest that P2Y14 in osteoblasts reduces cell sensitivity to mechanical stimulation and modulates osteoblast differentiation. I like to give the following comments.

  1. P2X receptors and UDP-sugars must introduce in clear, particularly the functional role.

Following the reviewer’s suggestion, we have now provided additional introduction to P2Y receptors (including P2X) and UDP-sugars.

  1. In bone, role of P2 receptor needs in vivo evidence such as the association with exercise and others.

We agree with the review that our conclusions related to P2Y14 involved in mechanotransduction are solely based off in vitro experiments and that in vivo validation is required. We have made this clear in the limitations section of the discussion and have outlined possible future directions to address this important question.

  1. a) Merit(s) of this study did not introduce in detail. b) Additionally, source of P2ry14 CRISPR/Cas9 plasmid and pharmacological inhibitors remained unknown.

  1. To date the role of nearly every P2 receptor has been studies in bone – P2Y14 was among the last that had not been investigated. This was the rationale for this study, which we have discussed in the introduction.
  2. We have now included additional details about the CRISPR plasmid construct in the generation of P2Y14 knockout cell line section of the methods. Sources for both the inhibitor and plasmid are provided in the reagents section of the supplemental materials – this is indicated in the reagents and solutions section at the beginning of the methods in the main text.

  1. a) Mild mechanical stimulus needs the reference(s). b) Pharmacological inhibition of P2Y14 did not show the same effect on cAMP as that in receptor knock-out cell-line. How to support the specificity of inhibitor?

  1. Our media dispensing method follows the approach taken by Kowal et al. (Kowel et al. (2015) Purinergic Signal. 11(4):533-550) who showed that dispensing 33% media volume evokes a mechanically-stimulated response in culture cells. This clarification is now included in our “Mechanical-stimulation” section of the Methods.
  2. Two possibilities for the inconsistencies between pharmacological inhibition and genetic knockout of P2Y14 are (i) off-target effects of Y14 inhibitor and (ii) off-target editing by Cas9. PPTN was identified in an unbiased screen of a small molecule library, and it has no discernable resemblance to other nucleotides suggesting that it is selective towards P2Y14 over other P2Y receptors. In line with this, an IP3 accumulation assay was previously used to demonstrate that PPTN has no effect on P2Y1, P2Y2, P2Y4 P2Y6 or P2Y11 (Barret et al. (2013) Mol Pharmacol 84:41-49). Therefore, if off target effects exist, they are not likely to be mediated through the P2Y receptor family. CRISPR/Cas9 gene editing can result in unintentional effects through off-target editing (Zhang et al. (2015). Mol Ther Nucleic Acids 17(4): e264). However, in our study we used a double-nickase variant of Cas9 that has been shown to have greater specificity and minimal off-target editing events compared to Cas9 (Ran et al. (2013). Cell 154(6): 1380-9). Taken together, the pharmacological and genetic methods used to perturb P2Y14 function in this study were relatively selective and unlikely to be the reason for observed inconsistencies. However, another possibility for these inconsistencies may relate to the duration of impaired P2Y14 function in the cells. With genetic ablation, impaired P2Y14 function was chronic while with pharmacological inhibition, it was acute. These differences in time scale may have been enough to achieve differential cell states that exhibited different intracellular cAMP characteristics. We have now included this perspective in the P2Y14 signal transduction section of the Discussion.

  1. P2Y14 seems involved in skeletal development but not for bone formation and maintenance. What is the role in functions?

We agree with the reviewer that this is an important question, however it requires further investigations, some of which are now outlined in the discussion.

  1. Role of P2Y14 receptor in murine osteoblasts has been characterized using the deletion and/or inhibition. What is the influence of overexpression?

Aside from the heterologous overexpression systems that were used to characterize P2Y14 signaling (identified in our meta-analysis; Table S2), we are not aware of any studies that have overexpressed P2Y14 in a system with functional P2Y14 expression. We have now provided additional discussion relating to this gap in knowledge in the limitations section of our discussion and suggest this as a future direction.

  1. It has been demonstrated that UDPG and P2Y14 inhibition had opposing effects on phosphorylation of ERK1/2 and AMPK. But UDPG may not be the main P2Y14 ligand for osteoblasts. How to explain this truth?

We have now included relevant discussion in the limitations section relevant to this question and propose that future studies confirming the release of UDPG from bone cells, as well as evaluation of UDP-sugars will be pertinent to determine what the main agonist of P2Y14 in bone is.

  1. P2Y14-inhibition results in higher sensitivity to mechanical and purinergic stimulation. This novel view lacks the mechanism(s).

We agree with the reviewer that we did not resolve the mechanism by which this sensitivity is achieved. We have identified this as a limitation and suggest that future studies investigate the link between P2Y14 and the cytoskeleton based on the functional coupling between P2Y14 and actin stress fibers reported by our study, and prior studies implicating the cytoskeleton in mechanically-induced signaling (references provided in main text; limitations section).  

  1. Limitation(s) of this report may assist the unclear point(s) from obtained results.

Following the reviewer’s suggestion, we have now included a limitations and future directions section in the discussion of the manuscript.